# Building an explanatory model for snakebite envenoming care in the Brazilian Amazon from the indigenous caregivers' perspective

**Altair Seabra de Farias[1], Elizandra Freitas do Nascimento[1], Manoel Rodrigues Gomes Filho[2], Aurimar Carneiro Felix[3], Macio da Costa Arévalo[2], Asenate Aline Xavier Adrião[1], Fan Hui Wen[4], Fabíola Guimarães de Carvalho[1], Felipe Murta[1,3,5], Vinícius Azevedo Machado[1], Jacqueline Sachett[1,3], Wuelton M. Monteiro[1,3]***

**1** School of Health Sciences, Universidade do Estado do Amazonas, Manaus, Brazil, **2** Distrito Sanitário Especial Indígena Alto Rio Solimões, Secretaria Especial de Saúde Indígena, Tabatinga, Brazil, **3** Department of Teaching and Research, Fundação de Medicina Tropical Dr. Heitor Vieira Dourado, Manaus, Brazil, **4** Butantan Institute, São Paulo, Brazil, **5** Leônidas e Maria Deane Institute, Fiocruz, Manaus, Brazil

* wueltonmm@gmail.com

**Data Availability Statement:** Data underlying the findings are fully available in the manuscript supplementary files.

## Abstract

### Background

In the Brazilian Amazon, snakebite envenomings (SBE) disproportionately affect indigenous peoples. Communication between indigenous and biomedical health sectors in regards to SBEs has never been explored in this region. This study aims to build an explanatory model (EM) of the indigenous healthcare domain for SBE patients from the perspective of the indigenous caregivers.

### Methodology/Principal findings

This is a qualitative study involving in-depth interviews of eight indigenous caregivers who are representatives of the Tikuna, Kokama and Kambeba ethnic groups, in the Alto Solimões River, western Brazilian Amazon. Data analysis was carried out via deductive thematic analysis. A framework was built containing the explanations based on three explanatory model (EM) components: etiology, course of sickness, and treatment. To indigenous caregivers, snakes are enemies and present conscience and intention. Snakebites have a natural or a supernatural cause, the last being more difficult to prevent and treat. Use of ayahuasca tea is a strategy used by some caregivers to identify the underlying cause of the SBE. Severe or lethal SBEs are understood as having been triggered by sorcery. Treatment is characterized by four components: i) immediate self-care; ii) first care in the village, mostly including tobacco smoking, chants and prayers, combined with the intake of animal bile and emetic plants; iii) a stay in a hospital, to receive antivenom and other treatments; iv) care in the village after hospital discharge, which is a phase of re-establishment of well-being and reintroduction into social life, using tobacco smoking, massages and compresses to the affected limb, and teas of bitter plants. Dietary taboos and behavioral interdictions (avoiding contact with menstruating and pregnant women) prevent complications, relapses,

**Funding:** J.S. and W.M.M. were funded by Conselho Nacional de Desenvolvimento Científico e Tecnológico (CNPq productivity scholarships). W. M.M. and J.S. were funded by Fundação de Amparo à Pesquisa do Estado do Amazonas (PRÓ-ESTADO, call 011/2021 - PCGP/FAPEAM, call 010/ 2021 - ÁREAS PRIORITÁRIAS, call 003/2022 - PRODOC/FAPEAM) and by the Ministry of Health, Brazil (proposal No. 733781/19-035). F.M. is funded by Fiocruz (Inova scholarship). F.M. was also funded via Programa Inova Fiocruz and VPAAPS/Fiocruz, project "Contribuição para o desenvolvimento de estratégias para o fortalecimento do SasiSUS, considerando as vulnerabilidades emergentes e reemergentes em saúde". The funders had no role in study design, data collection and analysis, decision to publish, or preparation of the manuscript.

**Competing interests:** The authors have declared that no competing interests exist.

and death, and must be performed up to three months after the snakebite. Caregivers are in favor of antivenom treatment in indigenous areas.

## Conclusions/Significance

There is a potential for articulation between different healthcare sectors to improve the management of SBEs in the Amazon region, and the aim is to decentralize antivenom treatment so that it occurs in indigenous health centers with the active participation of the indigenous caregivers.

## Author summary

The Brazilian Amazon is the region of the Americas that has the highest incidence of SBEs, and indigenous populations are disproportionately affected. In order for Brazil to achieve a reduction in mortality and disability from snakebite envenomations, it is vital to understand how indigenous and professional healthcare sectors interact in indigenous villages. The indigenous caregivers interviewed explained that the occurrence of a snakebite can be a natural event, usually due to the lack of attention of an indigenous person during their daily activities in the forest, or a supernatural event that was caused as a punishment or by witchcraft. The latter type of snakebites is more serious and it is difficult to intervene to promote healing. Indigenous caregivers recommend a series of rituals in order to protect an individual from snakebite or heal the already diseased body. In addition, a series of therapeutic resources derived from plants and animals is used for treatment and rehabilitation of the indigenous person. In this itinerary, there is no impediment on the part of the caregivers to include a stay in hospital in order to carry out the treatment with antivenom and other care deemed necessary by the doctors. This finding is crucial to improve the effectiveness of snakebite treatment in an integrated biomedical-indigenous model, thus reducing the contradictions and tensions present in the daily practices of both health teams and indigenous caregivers. The engagement with local caregivers to combine the indigenous health care model with a timely referral of SBE patients to a facility equipped with antivenom is a major determinant of success in the control of SBEs.

## Introduction

Snakebite envenomings (SBEs) represent a medical emergency that mainly affects populations living in underdeveloped tropical countries [1]. The highest incidence of SBEs is reported in the Brazilian Amazon, with a disproportionate burden for indigenous populations [2]. In this region, the case reporting system shows a 7.5-fold higher incidence in indigenous villagers (333.5 SBE cases/100,000 inhabitants) compared to the non-indigenous population (72.2/ 100,000 inhabitants) [3]. In addition, case-fatality rate from SBEs was significantly higher among indigenous villagers (1.4%) versus non-indigenous populations (0.5%) [3]. Furthermore, the frequency of late medical care is significantly higher in indigenous villagers [3,4]. Antivenom is not provided in indigenous community health centers, and transport to urban areas is needed to complete the therapeutic itinerary [2,3,5].

In Brazil, health care for indigenous villagers is assigned to the Indigenous Healthcare Subsystem within the scope of the Unified Health System (*Sistema Único de Saúde*; SUS). The organizational model is based on Special Indigenous Health Districts (*Distrito Sanitário*

*Especial Indígena*; DSEIs), which is a service organized to provide primary healthcare services to Brazil's indigenous communities within their ethnocultural settings [6]. In this system, the gateway for treatment for SBE patient is a community health center, which is able to offer only basic first aid, such as cleaning the bite site and analgesics. Patients are then transferred to the DSEI central health bases where a subsequent transfer to the reference hospital is obtained for antivenom treatment, and, if necessary, the treatment of complications arising from the SBE, such as secondary bacterial infection, severe local tissue necrosis and acute kidney injury [3].

The establishment of an indigenous healthcare subsystem has not been sufficient to fully guarantee quality of care and good health indicators in indigenous populations [7]. As noted for SBEs, other health problems also have a disproportionate burden among indigenous villagers, such as parasitic and infectious diseases [8–10], anemia [11] and undernutrition [12,13], in a worrying setting of mixed morbidity with obesity, diabetes and hypertension [14–16], alcohol abuse [17] and mental disorders [18]. In addition to a scenario of an inadequate healthcare coverage and geographic barriers, which noticeably limits access to health services [19], there is a lack of effective communication between health agents guided by the hegemonic ideology of biomedicine, and indigenous caregivers in the villages. This may generate resistance to or low acceptability of the health services on the part of indigenous people, as well as prejudice against ancestral healing practices on the part of health professionals [20,21].

Understanding the interaction between indigenous and professional healthcare sectors in indigenous villages is a crucial factor if one is to improve the global effectiveness of indigenous health care. This study assumes that the elaboration of an explanatory model (EM) for the indigenous healthcare model offers an arsenal of information for reducing contradictions and tensions still present in the daily practices of health teams at the local level when seeking to prevent and treat SBEs. The engagement with traditional caregivers to combine the indigenous healthcare model with a timely referral of SBE patients to a facility equipped with antivenom also depends on this knowledge. In this study, we aimed to build an EM of the indigenous sector of healthcare for SBE patients from the perspective of indigenous caregivers, in the region of the Alto Solimões River, in the western Brazilian Amazon.

## Material and methods

### Ethics statement

This study involves collection of data from indigenous populations and consent was obtained from indigenous leaders from each village. After this consent was given, the study protocol was submitted to the Health Research Coordination at the National Council for Scientific and Technological Development (COSAU/CNPq) and to the National Indigenous Foundation (FUNAI). Subsequently, with the approvals from COSAU/CNPq and FUNAI, the protocol was submitted and approved by the National Research Ethics Commission (approval number 4,993,083/2020). All participants signed a consent form after reading of the study objectives and procedures. To ensure their understanding of the study, the researcher responsible for the interviews was always accompanied by a native speaker of the participant's language.

### Study design

We conducted explanatory descriptive study in the Brazilian Amazonia to understand the role of the folk healthcare sector in treating SBEs in indigenous villages, within a cultural process that is managed primarily by indigenous caregivers. In-depth interviews were performed in the indigenous village with caregivers from January to December 2021, in the indigenous village. The study was conducted according to the Consolidated Criteria for Reporting Qualitative Research (COREQ) guideline (S1 File).

## Setting

The study was performed with indigenous caregivers living in the Special Indigenous Health Districts (SIHD) named Alto Solimões River, in the municipalities of Tabatinga, São Paulo de Olivença and Benjamin Constant, in the state of Amazonas, western Brazilian Amazonia. The SIHD is a healthcare administrative model that is responsible for providing primary health care to Brazil's indigenous communities [3]. The SIHD Alto Solimões River serves a total of 70,519 indigenous inhabitants that live in 231 villages, in 13 health basic hubs for indigenous health with a complete multidisciplinary indigenous health team (doctors, nurses, dentists, psychologists, pharmacists, laboratory technicians, nursing assistants, oral health technicians, indigenous health agents, and boat operators). In this SIHD, the indigenous people belong to seven ethnic groups (Tikuna–the most numerous, Kokama, Kaixana, Kambeba, Kanamari, Witoto, and Maku-Yuhup). The SIHDs do not provide antivenom treatment for SBE patients, and only perform the first aid via wound cleaning and analgesics. When an SBE is confirmed, the patient is transferred to the nearest hospital in an urban area to receive the antivenom [3].

## Research team and reflexivity

One male researcher (ASF) interviewed the participants. He is of indigenous origin (Kambeba) and has a background in nursing, with a Master of Science degree in Public Health, and is a specialist and professor of Indigenous Health. The script of the interview was designed by the research team with the support of a licensed nurse who is a member of the Tikuna ethnic group from the community of Feijoal, in the SIHD Alto Solimões River, municipality of Benjamin Constant; his father was an indigenous village chief and his grandfather a shaman. The study team also included one physician and three nurses with extensive experience in SBE research, one educator who is a specialist in public health and qualitative research, one nurse who is a specialist in indigenous health with experience in the SIHD Alto Solimões River (has witnessed several SBEs in the study area), and one epidemiologist.

## Survey participants

Indigenous caregivers over the age of 18 were invited to participate in the study. The selection of participants began with the discussions with the health managers and the multidisciplinary team of indigenous health workers of the municipalities, which together were defined as the caregivers working in the villages. In total, eight caregivers were included and interviewed in this research, these being five of the Tikuna, two of the Kokama and one of the Kambeba ethnic groups. Indigenous caregivers were selected from village actors with socially legitimate status, roles and power relationships with villagers who seek them out in case of health disorders, which were identified with the help of public authorities working in the indigenous district (Fig 1). The sample was of convenience, and all participants were indicated by the health professionals who work in the units of these villages. No refusals were observed. None of the participants had an established relationship with an author prior to study commencement.

## Data collection

An experienced researcher conducted the in-depth interviews using a semi-structured script (Table 1) in a quiet, comfortable space in the participants' homes. One silent observer from the research team was also present. Interviewer and observer introduced themselves to the participants at the beginning of the interview and gave a short personal background and explained their role in the study, as well the study objectives. The interviews lasted on average for an hour and were recorded via an audio recording device. The interviewer and silent observer

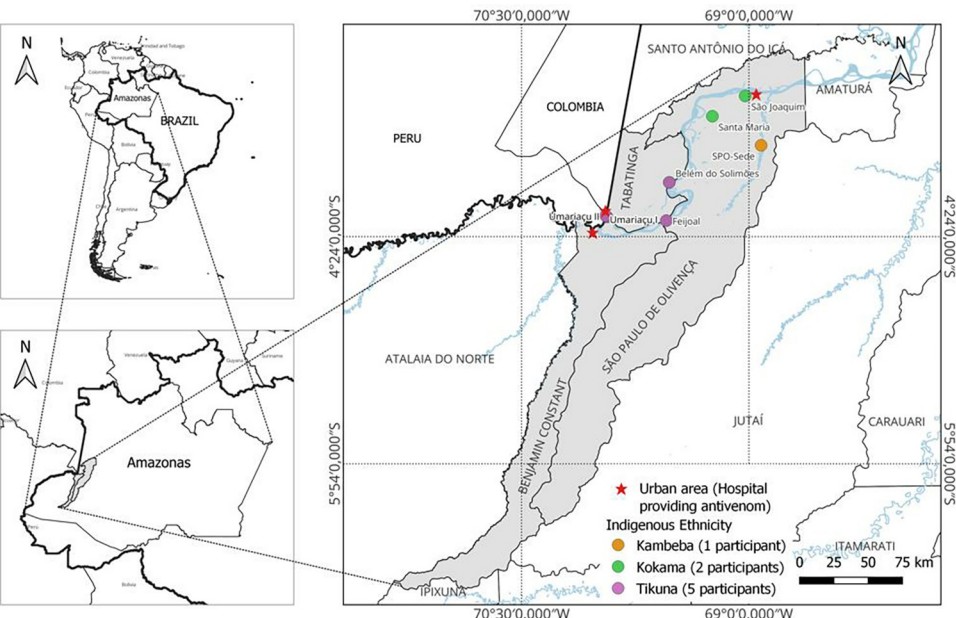

**Fig 1. Location of the study area, in the Special Indigenous Health Districts named Alto Solimões River, in the state of Amazonas, western Brazilian Amazon.** The location of the village of each caregiver is presented. The base used to create map is from the IBGE (Brazilian Institute of Geography and Statistics), which is freely accessible for creative use in shapefile format, in accordance with the Access to Information Law (12,527/2011) (https://geoftp.ibge. gov.br/cartas_e_mapas/bases_cartograficas_continuas/bc250/versao2019/).

**Table 1. Interview script for indigenous caregivers.**

| Question |
| --- |
| 1. Do snakebites occur in your village? |
| 2. How often do they occur? |
| 3. Have you ever seen snakebites occur? |
| 4. Why do snakebites occur in your village? |
| 5. Do snakebites have any meaning in your village? |
| 6. When you find a snake what should be done? |
| 7. When a snakebite occurs, what does the indigenous person feel? |
| 8. What can happen to the indigenous person who was bitten by a snake? |
| 9. What situations can aggravate snakebites? |
| 10. What should be done immediately after snakebite? |
| 11. What indigenous treatments are used for snakebites? |
| 12. When the indigenous person is bitten, does he/she have to go to the hospital? To do what? |
| 13. In which circumstances does the indigenous person need to go to the hospital? |
| 14. How long after the snakebite does indigenous person normally go to the hospital? |
| 15. What were the three most important snakebite cases you treated? |
| 16. What does an indigenous person do to avoid being bitten by a snake? |
| 17. How does an indigenous person protect the body so as not to be bitten? |
| 18. If there were boots and chaps in the village would an indigenous person use them to protect themselves from snakes? |
| 19. Is there any protective equipment used against snakebites that the indigenous person would like to use but they do not have in the village? |
| 20. To end the interview, can you give a summary of what you can and cannot do regarding snakebites? |

took field notes. The interviews were transcribed, de-identified, and loaded into MAXQDA 20. Transcripts are available in S2 File. No interviewer-related biases were identified.

Of the eight interviews, six were carried out in Portuguese, one was carried out with the help of an interpreter, and one was carried out with the support of the caregivers's daughter and granddaughter, who helped the participant to remember details of the SBE he had experienced.

## Data analysis

Interviews transcripts were analyzed using inductive content analysis. Data source triangulation was employed (transcripts, field notes) during the analysis. Two independent researchers (ASF and WM) created separate codebooks in MAXQDA 20 software, then discussed any discrepancies and established a common codebook for the analysis. The codebook developed was then applied to the interviews in an inductive content analysis [22]. Interview data was analyzed to understand indigenous caregivers' explanations in an analytical framework with three themes: i) Etiology; ii) Course of sickness (with three subthemes: onset of symptoms, pathophysiology, and severity and prognosis); and iii) Treatment [23]. A deductive layer of coding was applied after the initial inductive analysis to identify subthemes in the etiology and treatment themes, which complemented the EM structure for SBE management from the indigenous healthcare domain perspective. Boundaries and points of interaction with popular (patients' self-care and community support) and biomedical domains and the indigenous EM were presented. The analysis was concluded August 2022.

## Results

### Characteristics of the participants

A total of eight indigenous caregivers were included, with seven men and one woman, and ages ranged from 47 to 82 years. The participants belonged to the Tikuna (five), Kokama (two) and Kambeba (one) ethnic groups. Regarding religious affiliations, six are of the Saint Order Cross and two are Catholic. Table 2 describes the roles by which caregivers recognize themselves in their villages.

### Etiology

**Perpetrating snakes.** The indigenous caregivers are aware that SBEs are caused by the inoculation of a snake's venom into the body of a human being. There is consensus that the Amazonian pit vipers (*jararaca*, in Portuguese; *Bothrops atrox* L.) are the main causative agents of ing in the region. This snake generally bites the feet and legs. The bushmaster (*surucucu-pico-de-jaca*, in Portuguese; *Lachesis muta* L.) causes a lower number of bites, but it is much feared due to its aggressiveness, more potent venom, and long length, which can reach up to 3 meters. As it is very long, bushmasters bite "*only the buttocks and thigh of the individual*" (Participant 3). This snake is very heavy and strong and, when its lunges, it knocks the victim to the ground. The third most-aggressive agent is the two-striped forest-pitviper (*jararaca-verde* or *jararaca-papagaio*, in Portuguese; *Bothrops bilineatus* Wied-Neuwied), a snake that lives in the tree branches and bites the face of an individual harvesting açaí or hunting arboreal animals like monkeys.

Encounter with snakes during daily activities is an ordinary event for indigenous villagers, in particular for men, since thay have the social role of hunting, fishing and collecting fruits and materials for handcrafts and ornaments in the forest. To a lesser extent, women and children can also be affected when they help in the care of plantations. Due to this condition of

**Table 2. Characteristics of the participants.**

| Participant | Gender | Age | Ethnic group | Religion | Social role |
|---|---|---|---|---|---|
| P1 | Male | 50 | Tikuna | Saint Cross Order[1] | P1 prefers to be recognized as a healer, not a *pajé*[2]. He explained that a healer does not know sorceries, so he's not strong enough to fight evil, but he helps people, especially children. The healer knows how to use power because he takes away evil and protects himself. A healer knows medicine for all illnesses and pains, practices the good of healing all the time to help people. He emphasizes the use of dietary restrictions, the use of bitter tree bark and smoking using various substances as therapeutic resources. |
| P2 | Male | 47 | Tikuna | Saint Cross Order | P2 sees himself as healer and *pajé*. He has the gift of prayer to cure pain. He is a specialist in caring for indigenous children, especially in cases of vomiting and diarrhea. |
| P3 | Male | 78 | Kokama | Saint Cross Order | P3 sees himself as a healer whose specialty is the use of plants as therapeutic resources. He treats humans and domestic animals, such as dogs. |
| P4 | Male | 64 | Kambeba | Catholic | P4 recognizes himself as a healer. He resorts to using ayahuasca tea[3] to expand his mind and his ability to diagnose and cure illnesses. He uses chants, massages, and smoking with hawk's beak, feathers and talons as treatment resources. He has a special room to take care of sick people. |
| P5[4] | Male | 55 | Tikuna | Saint Cross Order | P5 recognizes himself as a *pajé*. He explained that he is seen in the village with fear and respect, for the power to dominate good and evil. He reported that he has difficulty passing on his knowledge to his children and grandchildren as they do not want to be associated with sorceries. |
| P6 | Male | 82 | Kokama | Saint Cross Order | P6 introduces himself as a shaman and has received healing power from God. His healing practices focus on the use of plants (ambé—*Philodendron imbe*, and cubiu—*Solanum sessiliflorum*, mainly). He works with three spirits. He worked as a rubber taper for 36 years and reported that he was bitten by snakes (jararacas) several times. |
| P7 | Male | 52 | Tikuna | Catholic | P7 recognizes himself as a *pajé*. In his childhood (5 years), the community recognized that he possessed healing powers. When he was 18 years old, he was taken from his village by a non-indigenous boss and brought to Manaus so that he could work in an *Umbanda*[5] house teaching shamanic practices. He was rescued by public authorities five months after his arrival. He uses vegetable latex to treat several illnesses. |
| P8 | Female | 52 | Tikuna | Saint Cross Order | P8 recognizes herself as a shaman, prayer and healer. She began her activity at the age of 16, having been taught by her father and grandfather. She has the gift of divination and the ability to control situations of conflict. She uses and recommends the use of amulets for protection against physical and spiritual ills. She wears a bracelet made with jararaca skin as an amulet in her everyday life. |

[1] The Saint Cross Order is a Messianic order that proliferates among indigenous communities in the Alto Alto Solimões region. It was founded by a missionary in Minas Gerais, southeastern Brazil, in the early 70's, who after traveling through several countries in South America, ended up settling in the area of this study. Currently, there are around 2,000 followers.

[2] *Pajé* is the term used to denote a shaman among Brazilian indigenous people, and serves as a counselor, healer, sorcerer, and spiritual intermediary of a community.

3 Ayahuasca is a psychoactive and entheogenic tea produced by combining the *Banisteriopsis caapi* vine with various other plants, in particular, *Psychotria viridis*. Its production and consumption are traditional in some Amazonian indigenous communities, as part of the folk medicine of these peoples. Recently, it has spread through Western society, and is used in rituals of different social groups and religions.

[4] This interview was fully conducted with the intermediation of a native speaker of the Tikuna language.

[5] Umbanda is a Brazilian religion that combines elements of Catholicism and Kardecism, the tradition of African *orixás* and spirits of indigenous origin. It was founded at the beginning of the 20th century in southeastern Brazil, in the city of São Gonçalo, state of Rio de Janeiro.

continuous and intense exposure, all caregivers reported having already treated cases of snakebites. Participant 6 has been bitten before. Snakebites happen more frequently in the rainy season, as the river level rises and snakes are forced to migrate to the sandbanks and upland areas where the plantations and houses are located. According to Participant 1, snakes are most active from 6 pm to 3 am: "*The snake also looks for food, right?*" But there is an exception, according to the same *pajé*, for pregnant snakes. These are active and very aggressive at any time of the day.

**Natural versus supernatural causes of snakebites.** Participants believe that the snake is an animal that has a conscience and a will of its own. Analysis of the interviews shows that SBEs have both natural and supernatural causes. The snake is "*an evil spirit*" (Participant 8). Snakes are harmful, dangerous, treacherous, and vengeful towards humans; thus, an enemy to be fought. A fair reason for snakes to attack and bite a person is having been offended or

disturbed in their natural space. An inattentive or careless person may step on a snake or get too close to it, and thus be attacked. It is very difficult to avoid this type of bite, "*unless you do not leave your house*" (Participant 5).

SBEs can also have a supernatural cause, in two ways:

i. Firstly, not behaving correctly in your community can attract a snakebite as punishment. When a person is not good, does not comply with the social rules of the community, is lazy, interferes with other people's work, then he will be bitten by the snake. Lying, and disrespecting the elderly can make the individual vulnerable to a snakebite.

ii. Secondly, if one is envied or hated by another person, this can make the individual a target for sorcerer-mediated spells. *Pajés* and sorcerers are distinguished in their social role by the purposes of their actions: while the *pajés* attribute to themselves the role of healing and helping the villagers, the sorcerers, in addition to their healing powers, accept performing activities that can harm other individuals. In this case, an indigenous person orders a sorcerer to arrange the death of his enemy or competitor via a SBE. This evil desire is moved by the envy of an individual who has more money or property, or who has won a prominent role in the village. "*This envious individual asks this of the sorcerer because he does not have the courage to do it himself*" (Participant 1). The snake does not receive a direct and in-person command from the sorcerer, but a spiritual order from the sorcerer's will, which drives the snake along a path focused on finding the victim. The snake in this case works as a vector of the evil feeling, chasing the target person until it fulfills the order received from the sorcerer. Sorcerers are very common in indigenous villages: "*in my village there are 72 sorcerers*", said Participant 1. However, most sorcerers play their sorcerer role discreetly in the village.

According to the participants, the *pajé* is the only person able to discover whether snakebite has a supernatural etiology. For instance, a *pajé* can use tobacco and ayahuasca tea in rituals with protective chants that involve community members. This ritual aims at expanding consciousness and clarifies the underlying real cause of the snakebite (Fig 2).

**Preventing snakebites.** A snakebite is an unpredictable event; generally, no one is walking through the forest and thinks they are going to be bitten. In addition, the snake camouflages itself very well, it knows how to remain invisible to humans in the midst of the foliage. Therefore, SBEs could be prevented only if individuals are very attentive and careful during activities in the forest —not because they could see the snake and keep a distance from the potential aggressor, but to kill the snake before the attack. All the participants said that killing the snake before being attacked would be the best way to prevent SBEs. Indigenous villagers carry a machete with them for their activities in the forest, and this same tool is of great use for killing snakes.

"*If the individual did not get too scared when he saw the snake, he can kill it. If he is with a companion, the companion can help resolve it*" (Participant 3).

"*The snake bites us. Kill it quickly! Because the snake is not friends with us. It is an enemy. You have to kill it!*" (Participant 5)

"*You must kill the snake! You won't leave a snake alive, right? I never leave it. . . When it runs away from me, I can't kill it. But if I can kill it, this very miserable creature will not escape*" (Participant 6).

When one has to keep one's mind occupied searching for food or other essential materials for survival, it is difficult to continuously keep your eyes on the ground. According to the

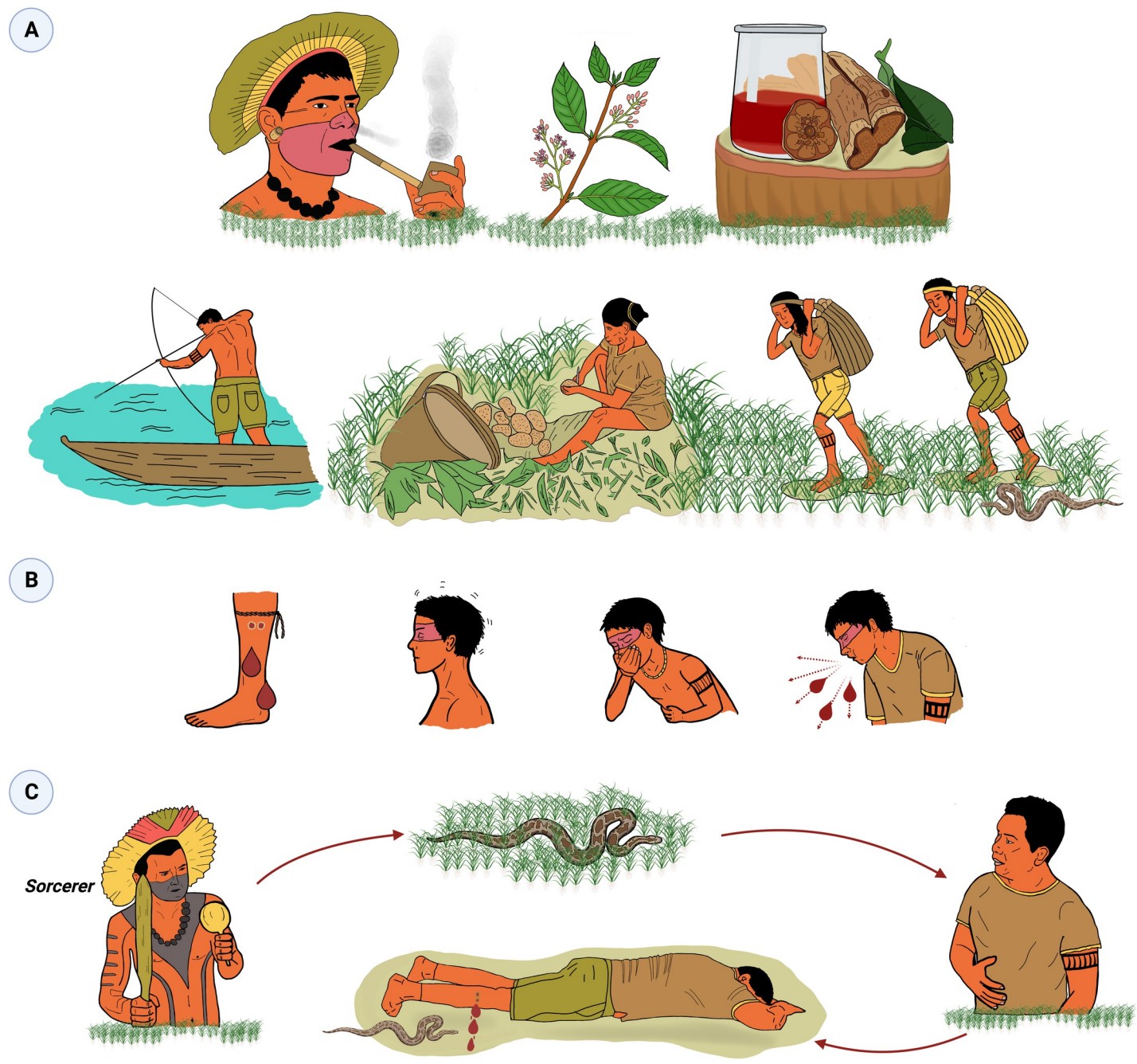

**Fig 2. Etiology of snakebites according the caregivers' perspective.** A) The *pajé* is the only individual able to discover whether a snakebite has a natural or supernatural etiology, based on rituals including tobacco and ayahuasca. B) The natural cause of snakebites is related to attacks and bites of snakes disturbed in their natural space. An inattentive or careless person may step on a snake or get too close to it, and be attacked during their work in the forest. C) The supernatural cause of snakebites is mediated by a sorcerer's spells. In this case, an indigenous person orders a sorcerer to arrange the death of his enemy or competitor via a snakebite envenoming.

participants, personal protective equipment such as boots and leggings are little used by indigenous people. According to them, they would accept using this type of protection if they received it. However, some quotes show that this equipment does not ensure total protection

from envenoming, since "*as the snake's fangs are very long, worse than an injection needle, if it bites, you have had it*" (Participant 3). In addition, "*even if the snake's teeth do not reach the individual's skin, the venom will contaminate the boot, which will be unusable and must be discarded*" (Participant 7).

Some rituals were mentioned by the shamans are performed to protect individuals against snakebites. These rituals are performed in the village with the assistance of an indigenous caregiver or by the individual or family members. The rituals are briefly described in the Fig 3.

In the case of SBEs caused by the work of sorcerers, only a *pajé* can identify the cause and reverse the situation. Participant 1 reports a case in which the target of the spell was his uncle, who was also a sorcerer. Sorcerers, being people that are feared and hated by their enemies, are frequent targets of sorcery. On a visit he paid to his uncle's house, he almost stepped on a small jararaca, but was not attacked. He was not attacked as the snake was not assigned to him, and was determined to find its target, his uncle. Upon seeing the urgent situation, he used a machete to kill the snake. He can only kill this snake used in sorcery because he was a good person. A person who was not good could also become a victim of the snake, which would continue to search for its target.

## Course of sickness

**Onset of symptoms.**   The snake is not always seen after the victim is bitten, so pain is the hallmark of snakebite envenoming. A numbness of the leg may occur, and the victim is no longer able to suspend this limb. The pain rises from the bite site and spreads throughout the body. Actually, pain in the affected limb is not even the main complaint immediately after the bite. Severe headache combined with dizziness is cited as a sign that appears quickly after the bite, as a result of the disorder triggered by the envenoming. The ill person may become lethargic, starts to moan and scream in pain, begin vomiting, can't walk or see, and may pass out abruptly: "*It looks like the person is dying* (Participant 1)". "*Usually they fall to the ground*", said Participant 2. Bleeding from the fang marks and from other orifices and spitting and vomiting blood are signs that appear quickly as well. Regarding the bleeding, Participant 6 said the following about a child he saw:

> "*Everything was swelling and bleeding. Blood from the mouth, nose, eyes, ears. Blood was coming out everywhere, from all the teeth. It didn't stop gushing blood. So everyone is already scared, right*?".

**Pathophysiology.**   From the indigenous caregivers' perspective, SBE pathogenesis is a process that starts with the injection of the snake's venom, which is responsible for the onset of the disorder. If the spread of the venom through the body is not promptly stopped, by a tourniquet for example, there is a progression of the illness, and systemic symptoms such as headache, dizziness, and bleeding will appear, with the possibility of death. The manifestations of the disease are associated with the rotting of the limb, which can progress to the bones and blood, until the person dies. If the venom penetrates the body too quickly, as in cases of sorcery, or if the patient takes too long to be treated, allowing time for the venom to spread, there is nothing else to do, and death will be certain. Envenoming caused by sorcery are more severe because the venom will be strengthened by the harmful elements invoked by the sorcerer, such as "*tobacco smoking spiritually manipulated*", explained Participant 1.

**Severity and prognosis.**   After the onset of symptoms, the main determining factor for the prognosis of the case is the underlying cause of SBEs. Except in cases in which the indigenous

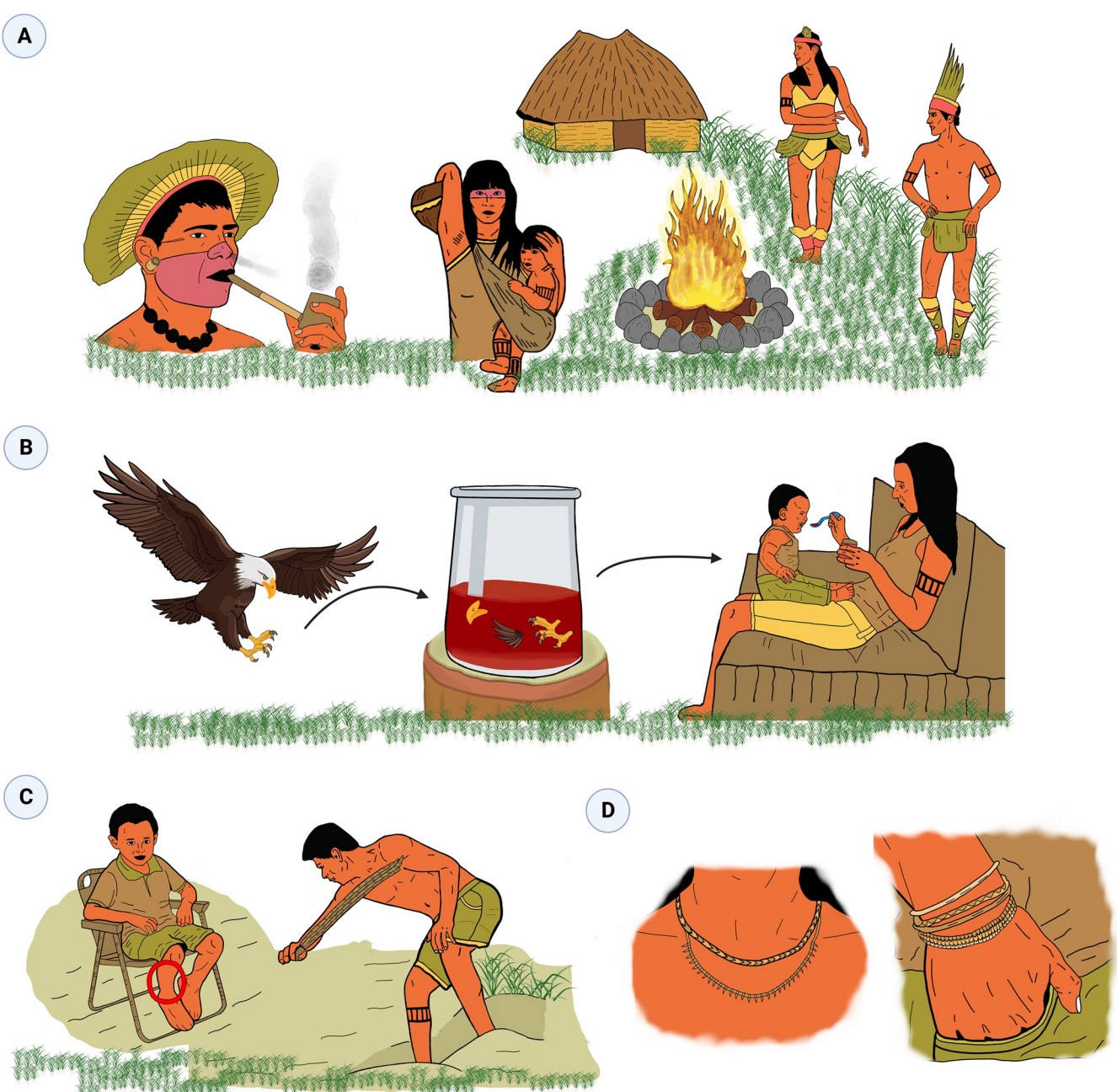

**Fig 3. Ritual performed by indigenous villagers of the Alto Solimões River area, western Brazilian Amazonia, for protection from snakebites.** A) Prayers, chants and smoking. Prayers and chants were widely cited by the indigenous caregivers as viable forms of protection, and can be done by anyone before entering the forest. "*During collective prayer rituals, snakes will reunite to know what is happening in the ritual, and understand that the people will be protected* (Participant 3)." However, if the villagers use prayers and songs as a joke, they will be harmed. Rituals of protection may also include smoke from the burning of dried tobacco or garlic leaves, and bee and wasp hives. In this technique, with prayers, chants and smoking, people and environments are protected against all illnesses, including SBEs. The procedure must be repeated by the indigenous caregiver from time to time to renew the protection. B) Rituals of 'acauã', the snake-eater falcon. The laughing falcon, also called snake hawk (*acauã* or *águia-cobreiro*, in Portuguese *Herpetotheres cachinnans* L.), is known for its characteristic song (changing from a joyful to a sad sound) and for feeding on snakes. Participant 4 explained this ritual. At the new moon, one of these falcons is killed, and its feathers, beak and talons are removed, burned and used to make an infusion. A spoonful of this infusion is given to a newborn baby. This baby will grow up protected from snakebites as will acquire the look of a falcon. As falcons have excellent vision for capturing snakes in the wild during a flight, snakes will not be able to resist the falcon's gaze, and will be chased away. The imitation of the falcon song by the indigenous people and chants about these snake predators are also used for protection, for yourself and for a loved one who is going to go into the forest. The falcons are recruited to take care of this person. C) Blessings of the legs using snake simulacrums. Some vines and bush stems have features reminiscent of snake skin. These simulacrums are used to beat the legs of individuals, especially children, who are going to start carrying out activities in the forest. This blessing promotes the protection of this child

throughout life. The ritual "*scares the snake away* (Participant 6)". "*The snake does not come close to him. When the jararaca looks at the person, it runs away* (Participant 8)". Participant 8 reports that she uses this form of protection for her children and husband, and recites the imperative "*Don't bite, snake, don't bite, snake, don't bite, snake*!", while performing the ritual. D) Amulets made from snake parts. Participant 4 explains that wearing a necklace with a bushmaster fang brings protection from evil and from the attack of snakes. The same *pajé* also quotes another amulet, a stone or pearl that is extracted from the inside of the boa constrictor's head. This stone has the power to hypnotize animals, such as birds and other prey, that would be attracted to the bushmaster's mouth. A human who carries this stone would have the same power as the serpent, and not recognized as a different one, i.e., a potential prey. Additionally, "*the amulet will increase the visual force of its carrier, saying 'you are mine' to the prey via thought* (Participant 4)". She explains that the bracelet works as a shield, and the *pajé* recognizes an evil person who comes close to her.

caregiver manages to intervene early in cases of sorcery, snakebite victims die instantly, or very quickly after being bitten, without time to be treated.

In parallel with the treatment, a series of dietary and behavioral interdictions are employed to prevent complications to the patients in the three months following snakebite ([Fig 4]). The indigenous caregivers understand that disobedience to these rules can aggravate the case, with the return of pain, swelling and bleeding, as the limb starts to rot again. In other words, relapses may result from the non-observance of these rules. Fishes are the main source of protein among indigenous villagers, and the exclusion of many fish species from the diet is what characterizes the major dietary changes to be implemented. Fishes with a long and fusiform body (*aruanã* or *sulamba—Osteoglossum bicirrhosum* Vand; *traíra—Hoplias* spp.), sharp teeth (*piranha–Serrasalmus rhombeus* Linnaeus), or with stingers (*mandi–Pimelodus* spp., *surubim—Pseudoplatystoma corruscans* Spix & Agassiz, *bodó*—family Loricariidae, *pirarara—Phractocephalus hemioliopterus* Bloch & Schneider, *piramutaba—Brachyplatystoma vaillantii* Valenciennes), cannot be eaten. The prohibition can also be due to other characteristics of the fish, such as aggressive habits, feeding behavior, smell, or other anatomical characteristics. Other fish species prohibited for SBE patients are *pacu* (family Myleinae), *piau* (*Leporinus friderici* Bloch), *pirapitinga* (*Piaractus brachypomus* G. Cuvier), *pirarucu* (*Arapaima gigas* Schinz), and *branquinha* (*Psectrogaster* spp.). Participant 2 explains the following: 1) Piranha, with its sharp teeth, "*eats the flesh from the inside, inflaming the flesh inside the wound. Then, the leg swells*"; 2) Catfishes with their stingers "*look like they're puncturing the flesh, then it starts to inflame as well*"; 3) *Sulambas*, as well as the snakes, are descendants of the *Cobra Grande* (Big Cobra), a mythological creature that is an enemy of humankind. Fishes that can be eaten without restriction are sardines (*Triportheus* spp.) and *curimatã* (*Prochilodus* spp.). Pork and tapir meat, very salty or fatty dishes, are also prohibited. The meat of venison and wild birds in general are allowed.

Contact with a woman who is menstruating or who has recently had sex with her husband, and pregnant women is totally contraindicated for snakebites patients up to three months after the bite. The rule extends to contact with the husband who had intercourse with his wife and to the husband of the pregnant woman. This includes being taken care of by her, getting close to her, and listening to her voice, looking at her or being looked at by her, or eating a meal prepared by a woman in any of these situations. Depending on the indigenous caregiver, this rule is valid for the entire family of the patient. The outcome for patients who fail to comply with this rule is the worsening of the case, and culminates in death:

"*A man was bitten by a snake, he was already well, 30 days after the bite, but suddenly a woman* (referring to a pregnant woman*), he didn't even look at her, he only heard her voice when she passed by. Not five minutes later, the man started screaming. This worried us a lot. As always, at the time, we worked very hard, and we had all the preparations, we had the preparations: feather, everything from the hawk, we went to arrange things and drink this vine (*referring to ayahuasca tea) *for the ritual, but he died* (Participant 4)."

"*When the woman is pregnant, don't go near her. People die! You've got to stay away. If it's five meters away... If the woman gets close to him, blood starts coming out of his teeth, mouth, sometimes nose, then it's worse*" (Participant 5).

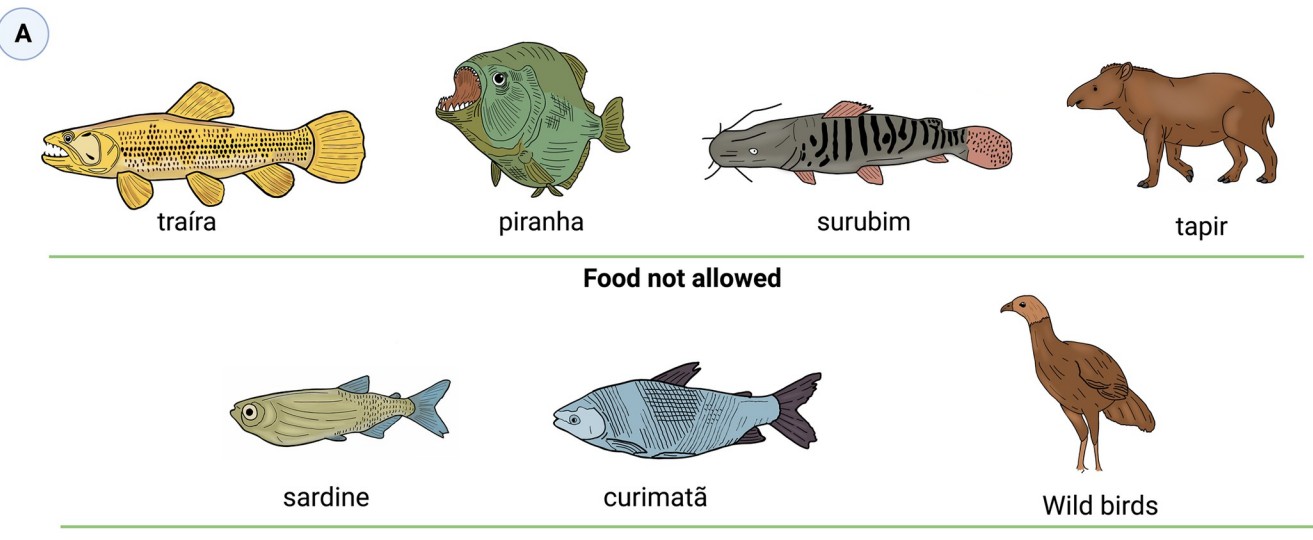

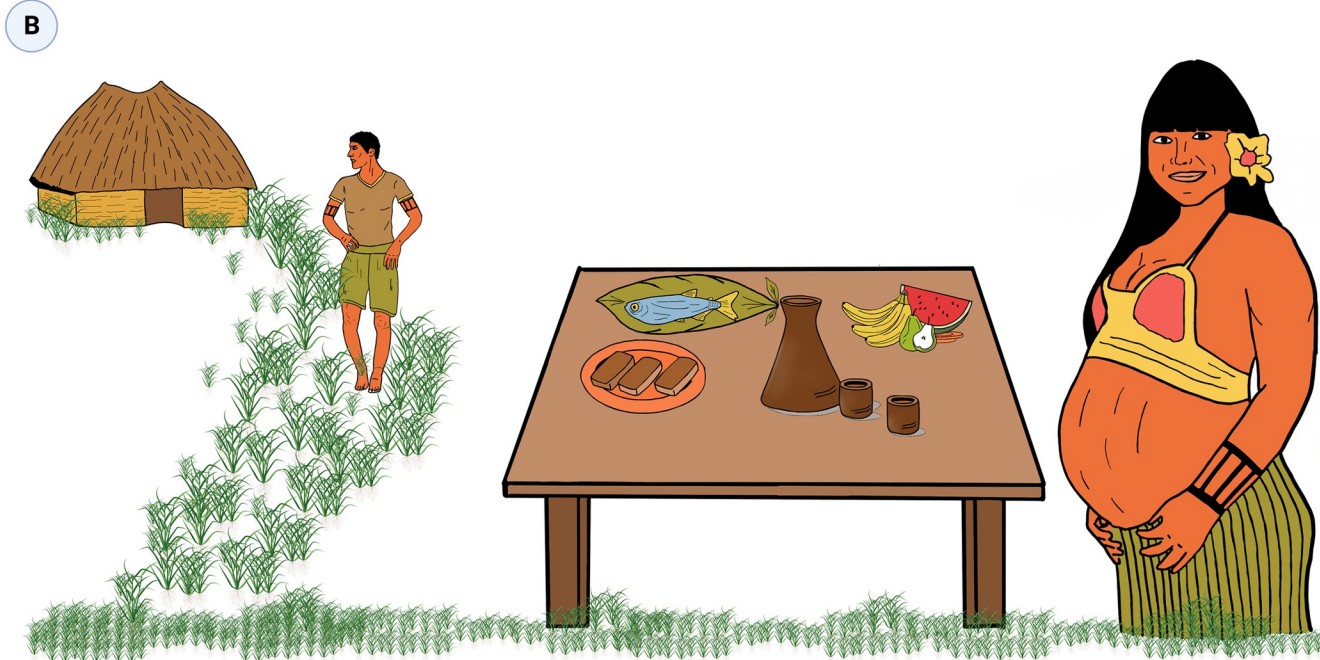

**Fig 4. Some dietary and behavioral interdictions used to prevent complications arising from snakebites.** A) Fish are the main source of protein among indigenous villagers, and the exclusion of many fish species from the diet is what characterizes the major dietary changes. Consumption of these forbidden foods by a snakebite victim, causes rotting of the affected limb, increases pain and swelling, and restarts bleeding, which can lead to limb paralysis, amputation, and even death. Fishes with a long and fusiform body (*traíra*—*Hoplias* spp.), sharp teeth (*piranha*–*Serrasalmus rhombeus* Linnaeus), or those with stingers (*surubim*—*Pseudoplatystoma corruscans* Spix & Agassiz), cannot be eaten. Tapir meat is also prohibited. Sardines (*Triportheus* spp.) and *curimatã* (*Prochilodus* spp.) can be eaten without any restrictions. The meat of wild birds in general are allowed. B) Contact with pregnant women is totally contraindicated for snakebites patients. Thus, it is common for indigenous people bitten by snakes to move away from the community to avoid contact with women in general, and even avoid eating a meal prepared by a pregnant woman.

Participant 7 explains about the danger of a pregnant woman who has contact with a SBE patient. In this case, the 'little animals' (he says that these little animals are the snake's off-spring) that are in the envenomed patient pass to the pregnant woman, and she starts to have

intense pain. The indigenous caregiver will need to intervene in these cases, and the ritual to reverse the disease in this pregnant woman lasts a whole day.

Lastly, snakebite patients should not have a sexual intercourse during the three months of treatment, due to the risk of the same complications as mentioned above.

**Treatment.** SBEs generally occur in the forest, during fishing, hunting and forestry, and affected individuals require an immediate transfer to the village. In the village, the therapeutic itinerary may be planned depending on the patient's severity and the perception of the different actors, and transport to the hospital is started whenever possible. In the hospital, the patient will receive medical care, especially antivenom treatment, and then return to the village to continue the treatment with the indigenous caregiver (Fig 5).

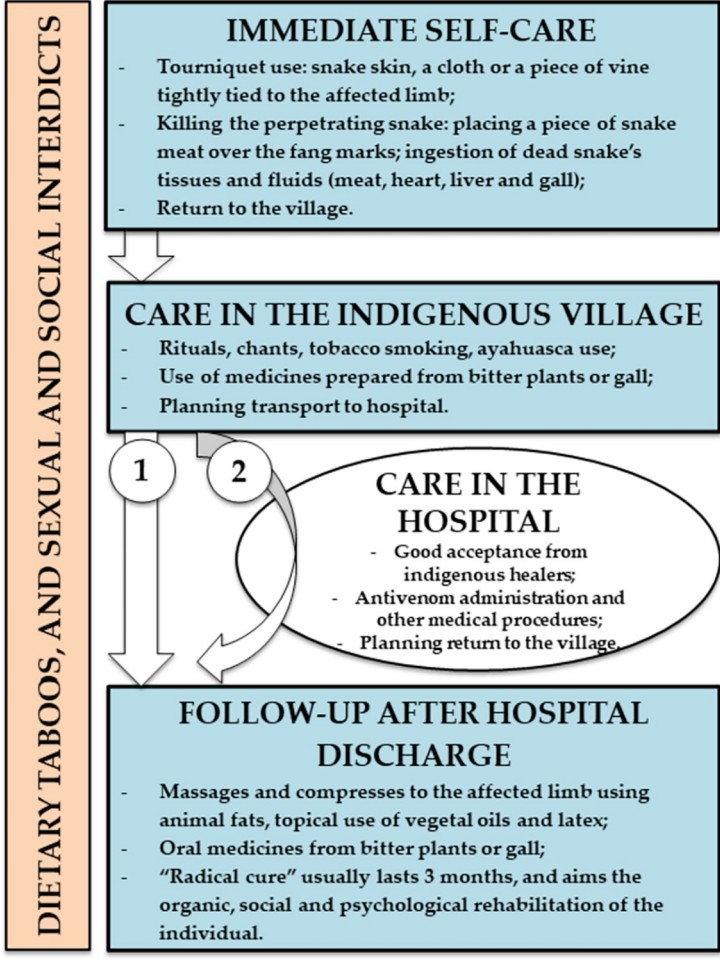

**Fig 5. Therapeutic itinerary of an indigenous snakebite patient from the perspective of the indigenous caregivers from the moment of the bite to rehabilitation.** Snakebite envenomings occur mostly in the forest, during fishing, hunting and forestry. At this time, self-care practices include the use of tissues of the dead snake that caused the injury. Bitten individuals require immediate transfer to the village, and rituals and plant-derived medicines and bile are part of the therapeutic arsenal. In the community, the therapeutic itinerary may be planned depending on the patient's severity and the perception of the different actors in the village. In some cases, there may be a refusal to go to the hospital, or there are no means of transport available (1). However, transport of the patient to the hospital is done almost whenever possible (2). In the hospital, the patient will receive medical care, especially antivenom treatment, and then return to the village to continue the treatment with the indigenous caregiver in the rehabilitation phase. Dietary taboos, and sexual and social interdictions are noticed throughout this itinerary.

**Immediate self-care after SBEs.**    After an SBE, some therapeutic strategies are available for self-care such as the use of tissues from the perpetrating snake itself, which is killed whenever possible. A piece of snake meat can be placed over the fang marks to "*suck out the venom and minimize pain*" so that the patient can be transported to the village. If the venom that the snake carries "*doesn't hurt for her, it doesn't hurt for us*", explained the shaman Participant 8. Still using the dead snake's body, its skin can be tightly tied to the affected limb, making a tourniquet, with the aim of preventing "*the venom venom from spreading up into the body*". Alternatively, bitten individuals can use a cloth or piece of vine collected locally to make a tourniquet. The ingestion of the dead snake's tissues and fluids (meat, heart, liver, tail tip, and bile) is also a treatment strategy that can be used by the bitten individual. To go out hunting or fishing "*you can also take with you painkillers such as metamizole and paracetamol and the bile of animals*" (Participant 1).

Participant 1 recommends:

"*Kill that snake. Then you take a little bit of his meat, a little bit of his meat, then you put it where the fang is* (referring to the bite marks), *you know? Then, this meat sucks. . . its venom. Sucked, sucked, sucked, sucked like this. Put a little on. Then, with a little, it stopped, right, the pain*".

"*Take off the skin. Tie it where, where is the pain like this, its skin. Take it off and tie it so it doesn't spread up* (the venom) *into your body, like this. Tie it tight. Tie it.*"

"That solves it."

"*And if, if you can stand it, to stand it, to stand it* (when arriving in the village), *take a little more and eat the meat*".

"That's to stop the pain."

After the primary care, the individual should make an effort to get to the village as quickly as possible. At this moment, his companions are recruited to help him make the journey. Participant 3 emphasizes that "*it is important that hunters always walk in twos because, if something happens to the partner, there is the other one to help, to carry him*". Participant 5 confirms this by saying that "*if the guy gets sick when he is out there in the woods. . . in the igapó. . . the guy has to shout for others, for other people to go look for him. . .*"

**Care in the village.**    Severe SBEs can be transported straight to the hospital, without care by the indigenous caregiver. However, if there is time, the indigenous caregiver can employ several treatment strategies, including rituals and medicines. As SBEs can have natural or supernatural causes, it is necessary that the treatment is comprehensive to contemplate these two possibilities. To decide on the best therapeutic strategy for a specific patient, the indigenous caregivers can use the ayahuasca tea with chants involving the villagers. Clarifying the underlying cause of SBE, such as the orders of sorcerers to kill an individual by snakebite, is key for therapeutic prescription.

"*Then we went to get the plants* (referring to ayahuasca). *There, women and men singing all the rituals of nature. . . You won't hear any kind of negative things because nature will rebuke evil, because we fight against evil. That is our goal.*" (Participant 3)

An important therapeutic resource used by the *pajés* is the use of tobacco (*Nicotiana tabacum* L.) by the smoking technique, in which the indigenous caregiver exposes the patient to the smoke generated by the controlled burning of the dried leaves of the plant. Smoking aims to "*to suck out the snake venom*" and remove disturbing fluids from the patient, their cohabitants, and places, and to attract beneficial energy from the natural elements, reversing them for the good of the patient. In this ritual, tobacco may be combined with other substances, such as the hive of the *jandaíra* bee (*Melipona interrupta* Lat.), *breu-branco* resin or oil (*Protium heptaphyllum* March.) and *cachaça* (a distilled spirit made from fermented sugarcane juice). Smoking combined with prayers is part of the healing repertoire that will continue until hospital

discharge and patient's reintroduction into the community, as the SBE can be the result of the evil eye, envy and spiritual aggressions.

A Kokama caregiver (Participant 3) explains the carrying out of a ritual to treat snakebite patients, though one that is no longer used. In this ritual, the victim and other villagers paint themselves with natural paint made with *korimã* (*jenipapo*, in Portuguese; fruits of *Genipa americana* L.). Community members carve pieces of wood into the shape of snakes. The *pajé* will then announce that the next day the patient will be able to walk. On the patient's face, a figure is drawn, in the shape of an S, representing a snake. In a kind of theatrical staging, tree branches are used to beat wooden snakes. The *pajé* emphasizes that at that time "*the snakes will be afraid*". Everyone participates singing and walking through the woods on the banks of a river. Everyone watches the victim bathe in the river: the venom from the patient's body is transferred to the water. Venom will thus be diluted in the river and carried away by the current. At the end of the ritual, the victim comes out of the water happy and greets the people who watched him: "*God will have watched over the sick person*".

In this phase, the study participants quoted as medicines to treat snakebites the i) ingestion of a few drops of bile from the *surucucu* (bushmasters, *Lachesis muta* L., or adult Amazonian lancehead, *Bothrops atrox* Wagl.) or lowland paca (*Cuniculus paca* L.); ii) ingestion of a preparation based on the wall of anthills, crumbled and mixed with the sap of young leaves of the açaí palm (*Euterpe oleracea* Mart.) and capeba leaves (also called pariparoba, *Piper umbellatum* L.); iii) ingestion of infusions of copaiba (*Copaifera langsdorffii* Desf.) and ginger (*Zingiber officinale* R.). Some of these oral preparations have emetic properties, and, according to the pajés, they act by expelling the venom from the patient's body.

**The role of the hospital.** Nowadays, some SBE patients, especially the most severe ones, are sent straight to the hospital. Some villages have community health centers, with indigenous health agents who will provide first aid to patients, administering painkillers, and arranging transport to the hospital in the city. The referral of the patient without the assistance of the indigenous caregiver is understood in different ways: some who live in areas closer to the cities, in greater contact with non-indigenous society, no longer seem to worry about it, while others resent the fact of this structural aspect of the culture being lost. However, none of the caregivers are opposed to patients seeking medical treatment in the hospitals. Participant 4 explains that the patient must be referred to the hospital "*for his safety, that many times, he can have other problems, some organic problems, like that, in the body. So, he can have a problem and in that problem, he can have low immunity, he can get worse and not resist death.*" Moreover, antivenom is cited as the treatment developed by doctors against snake's venom present in the patient's body.

**Care when returning to the village.** Hospital treatment is not understood as the final stage of treatment for snakebites. A re-establishment of a sense of well-being by the reintroduction of the individual into social life is planned upon the patients return to the village This phase of "radical cure" of the envenoming usually lasts 3 months, and aims at the organic, social and psychological rehabilitation of the individual. The indigenous caregiver seeks resources in rituals, the same as described above (tobacco smoking and ayahuasca) and new classes of medicines.

At this time, the application of massages and compresses to the affected limb has a prominent place in the patient's recovery. Procedures include massage with animal fats (guariba monkey, jaguar, alligator, stingray, anaconda), topical use of copaiba oil, and bandages with latex extracted from the sandbox tree (*assacu*, in Portuguese; *Hura crepitans* L.), guariuba (*guariúba* or *capinuri*, in Portuguese; *Clarisia racemosa*, Ruiz & Pav.), and milktree (*pau-de-colher*, in Portuguese; *Tabernaemontana echinata* Vell.).

Medicines for oral or topical use made from bitter plants are also indicated at this stage. The sap of the leaves of cubiu (*Solanum sessiflorum* Dunal) and capeba (*Piper umbellatum* L.) were the most widely recommended. The sap obtained from the leaves *Kalanchoe pinnata* Lam. (*língua de pirarucu* or *corama*, in Portuguese), *Justicia acuminatissima* (Miq.) Bremek (sara-tudo, in Portuguese), *Inga* spp. (inga tree), and *Echinodorus grandiflorus* Cham. & Schltdl. (chapéu de couro, in Portuguese) were also quoted. Infusions prepared with the bitter bark of the mango tree (*Mangifera indica* L.), yellow mombin tree (*Spondias mombin* L.), and orange tree (*Citrus sinensis* L.) are also on this list.

Participant 2 explains the mechanism of action of bitter plants as follows:

*"It's her sap, because it's strong, it's a pungent taste, it's bitter. . . That's it, with that it heals, it cuts the effect of the snake's venom. After the person takes the snake antivenom, then it* (bitter plant) *helps combined with the antivenom"*

"It heals the bite site."

"With that, it soothes the pain."

"Soothes the pain. That's why we use these plants up there."

Some medicines require more details regarding the technique involved in their preparation, as participant 3 explains about the cubiu leaves:

"Take three cubiu leaves. Then, you knead them well knead, extract the juice, strain it well, and put it in a liter of water. Give the patient about three glasses. It doesn't matter that it has already been four days, five days from bite. So, he/she will drink the juice. We sweeten it with very sweet sugar and then you put oil in it, food oil or electric oil, well-seasoned with oil. Then, it's ready. Then you will give it to the patient who was bitten by a snake. If he's about to die, but if God allows. . . But if you drink that. . . Even if you've already taken snake antivenom. . ."

The same participant mentioned some cases in which he successfully used the cubiu medicine. In one of them, a young man was bitten on the leg by a snake, and was hospitalized for five days in the city. According to the caregiver's report, the doctor released the man "*even in a severe condition because there was nothing else to do in the hospital*". Then, the man returned to the village, silent and quiet as a result of the SBE: the boy's father said he did not know "*if he was alive or dead*" when he sought help from the indigenous caregiver. Although the "*leg was already rotting and purple*", after starting treatment with the medicine, he felt fine, the pain stopped, the swelling subsided and he stopped coughing up blood. According to the *pajé*'s prediction, after three days, the young man would walk again. And so it happened: on the third day the boy started walking slowly and went out to urinate in the backyard of the house. "*Only through God was it possible for him to get well*", as the indigenous caregiver said.

## Discussion

### Indigenous caregivers and their role

The way in which the participants see themselves in relation to their healing practices and recall the constitution of knowledge and skills linked to them, represented two relevant topics in this study for the understanding of the uses of indigenous medicine in the treatment of SBEs in the villages where they circulate.

As for the first topic, participants recognize themselves as healers, *pajés*, prayers, or practitioners of natural medicine. In their quotes, the definitions of these terms present a polysemic condition, which results from variations in attributions, knowledge and skills corresponding to healing and protection practices, even within the same ethnic group. In addition, positive and negative connotations were attributed to them, depending on the value of the attributions, the talent to master healing and protection practices and the good or bad handling of supernatural forces. Considering the Tikuna ethnicity, according to Participant 1, the attributions of a

healer and a *pajé* are more relevant to the extent to which they can cure and protect people and combat witchcraft. A prayer, on the other hand, although he can practice healing and protection, has less strength to fight sorcery because he is not able to discover whether the SBE has a supernatural etiology. Participant 8 associates attributions of healing, shamanism and prayer. In her narrative, this action, as well as the volume of demands for her care, is perceived by villagers as a demonstration of her talent to master healing and protection practices. Regarding the management of supernatural forces, Participant 1 presented the figure of the sorcerer in opposition to healers, *pajés*, prayers, and practitioners of natural medicine. He reported that a sorcerer knows how to harm someone by wielding supernatural forces, such as spirits, and that he can cause an SBE by using tobacco to govern a snake's behavior and potentiate its venom. Finally, it is worth noting that Participant 5 stated that in his village the *pajé* is seen with fear, distrust, and prejudice, similar to the figure of the sorcerer.

Regarding the second topic, it is worth noting that this plurality of identities is reiterated in the narrative of the indigenous caregivers about the way in which they constituted themselves as healing agents throughout their lives and in the multiplicity of their knowledge and practices. Considering the Tikuna caregivers (Participants 1, 2 and 8), and the Kokama caregiver (Participant 3), it was observed that healing abilities were perceived by themselves and family members already in childhood. Knowledge in this regard, therefore, was transmitted gradually by older family members, following the logic of a dispersed learning that is subject to everyday events. Participant 8, however, indicated the increase in a learning agenda, since in her early youth she would have spent two years accompanying her maternal grandfather, also a caregiver, in order to learn the practices. From the circumscription of these milestones in the course of the lives of the indigenous caregivers (the shared perception of healing skills and the time taken to acquire knowledge for the practices), it can be inferred that the caregivers are also political agents in their villages. In other words, authority as healing agents was produced in the narratives and interpersonal relationships. According to Participant 2, in his first performance as a caregiver, he was sought out by a mother with a sick child. He asked her three tim"s: *"Do you believe me*?*"* And he added: *"If you believe me, I'll make your daughter well".*

As for the multiplicity of knowledge and skills that constitute healing practices, the variety of elements used in medicines and rituals, differences in food taboos and prohibitions, and cultural hybridity are notable. In his rituals, Participant 1 makes medicines from the bitter barks of trees and fruits, blows tobacco cigarette smoke to expel the snake's venom and clean the wound, and repeats prayers. In the treatment of the sick person, he indicates the prolonged adoption of food taboos and sexual prohibitions, as well as the application of drops of bile directly on the wound to relieve the pain. He uses smoking to also promote protection of the indigenous person. Participant 2's healing ritual also involves prayers and smoking. In the treatment, the tongue of the pirarucu and bitter plants are used to make medicines. He indicates food that should be avoided, but does not recommend prohibitions of any kind. Although he uses prayers, blowing smoke to protect the body from illness, he reported that there are no supernatural ways to protect yourself from snakebites. The most effective way would be to avoid going out at certain times and paying attention to where you step.

Participant 8, like the two aforementioned participants, uses copaiba leaves and ginger tea in her rituals, as well as Catholic prayers and blowing on the wound with tobacco to relieve the pain. She determines food taboos and prohibitions related to pregnant women. Participant 3 indicates that the best protection against snakebite is to kill the snake. In her rituals, he uses the cubiu leaf, prayers and chants of her ethnicity. In addition, a role-play in which all villagers participate can also be added to the healing ritual. Finally, it is worth mentioning cultural hybridity. In the narratives of all the participants, it was noticed that the healing practices related to indigenous medicine are influenced by previous contact of the interviewees with

non-indigenous people. And, although practices related to Catholicism were narrated more often, others were mentioned, such as divination through prayers and the use of ayahuasca tea, which was defined by Participant 1 as a drink that reveals unknown things.

### SBEs among indigenous villagers, a breakdown of the physical and social bodies

The few ethnographies available report an ambiguous relationship between indigenous people and snakes. Among the Tikuna, the great snake (*Yewae*) is the possessor of the fish tree (*Ngewane*), an enchanted tree that exists since the beginning of the world. When the weather is rainy and windy, the leaves of this tree fall and small eggs, similar to frog spawn, begin to appear on its trunk. The eggs will metamorphose into caterpillars. The caterpillars grow and descend to the trees' roots because of the lightning and thunders. The rain increases and the caterpillars come out transformed into various types of large and small fishes. The fish spread through the waters and take over water streams, lakes, igapós, and rivers, serving to feed people [24]. Similar myths are apparently shared by different peoples of the lowlands of the Amazon. The Baniwas, in the Negro River region, for instance, contend that fish and snakes are descendants of the Big Snake: fish and snakes are both enemies of humanity (*walimanai*), but people also depend on them for food. The creator god (*Niãpirikoli*), fights against snakes, killing them or expelling them from the lakes to provide space for humans. While defeating snakes facilitates human access to fishing sites, it simultaneously reduces the abundance of fish [25].

This constitutive association between fish, an essential food, and snakes, an enemy, marks the constant pursuit for balance between the different natural beings. The disrespect to this order affects humans in a harmful way, and they can be threatened by other humans, such as enemies directly or through sorcerers, or by non-humans, such as snakes, and extra-humans, such as the spiritual entities of the forest. The causes of SBEs vary in the explanations of the *pajés*, and even within the same ethnic group, and can be interpreted as an accidental encounter with an aggressor agent, or as a consequence of breaking with traditions, which makes the individual more susceptible to the snake's attack. In the words of some caregivers, this increased risk of being bitten is explained as a form of punishment. Finally, the snake's attack can be an effect of sorcery that is directed at an individual for various reasons, such as when their prestige increses in the village, if they accumulate properties, and in case of disagreements. SBEs as a punishment for inappropriate behavior or caused by supernatural causes, such as spells, has also been reported by traditional caregivers in Ghana [26]. In addition, the belief that SBEs caused by a curse or witchcraft are more lethal than natural SBEs has also been observed in traditional caregivers in Ghana and Eswatini [27,28].

Indigenous villagers' experiences of SBEs are not just a disturbance of their body physiology. The transition from a healthy state to an illness involves a biological change, but also changes the social status and identity of the patient and the community. More than just an unlucky encounter with an aggressive venomous agent, SBEs are part of the indigenous imagination, and require a series of collective obligations from villagers, from childhood and throughout life, for prevention and treatment. Thus, the rotting of the body that is caused by an SBE, which can generate disabilities and death, is the outcome of the disrespect of traditions that organize the social and political life, and include social and environmental interdicts, and dietary taboos. The fulfillment of the aforementioned precepts keeps the individual in harmony with the social and political organization and enables the indigenous person to return faster to their role in the village. All prevention and treatment strategies are based on maintaining or restoring the balance of the sick person, because the process of healing the body to return to collective experiences is a goal that depends on the commitment of many people. In

the EM presented here, the SBE is an 'illness', not a 'disease', from the perspective of the *pajés* and indigenous villagers. According to Kleinman [23], disease is a health problem from the biomedical perspective, and is reconfigured only as a change in the biological structure and functioning. This actually occurs in the *pajés'* comprehension, and local effects and systemic manifestations resulting from unclottable blood are understood as the outcome of the venom spreading through the body. However, the way of interpreting SBEs by the indigenous caregivers evokes a broader experience of symptoms and suffering; to the sick person and the members of family and villagers, the 'illness' is a breakdown of social 'body'. During the illness, the sick person is temporarily exempt from performing 'normal' social roles (such as hunting or fishing) but is expected to see being sick as undesirable and so they are under the obligation to try and get well as quickly as possible, cooperating with the advice of the indigenous caregiver in order to get better by complying with the social, environmental, and dietary interdicts.

## Body construction, prevention and treatment of SBEs

For indigenous societies, the body is built in relation to the environment and to other villagers. The body is the synthesis of natural elements and an equalizer of substances, in a constant search for balance, in a structure that can only be understood within the cosmology of these peoples. As the boundary of the body is neither clear nor constant, several disturbances by natural and supernatural aggressors transform this body, leading to illness. Thus, the prevention of an SBE should be interpreted as keeping indigenous individuals within the limits of the human condition, in a continued effort, which requires great caution in performing daily activities, food and sexual interdictions, and participation in rituals. Body transmutation, continuously invoking or permanently incorporating components from nature's entities, explains the mechanism by which preventive practices against SBEs work. In this sense, if it is possible to incorporate components of the body of a snake predator, such as a falcon, into the human body, protection will occur through the ability of these individuals to have more accurate vision in order to circulate in the forest, and to scare away snakes, which will see the indigenous person as a potential risk. Another strategy is to incorporate characteristics of the snake itself by using amulets made from snake parts, or by participating in rituals performed with snake simulacrums. Apparently, the physical characteristics and malice of a snake merged into the human body will benefit the individual and protect him in his activities.

In the case of either deaths or worsening, persistence or appearance of new symptoms, the ultimate cause of the SBE will be reviewed by the caregiver in an attempt to relate the patient's particular illness to his physical and social environment. Sorcery can be identified in this way, but undoing the spell may not always be possible. Sorcerers use various forms of supernatural techniques to bring misfortune to others, for which the specific invisible cause is not always discovered by the *pajés* in order to intervene on behalf of the victim, to eliminate the cause by counter attack. At this level, the cause of the disease is divorced from the snake's position in the ecosystem, from a western scientific perspective, and prevention will focus more on cosmological or social causes. However, preparing the body of these individuals to make them less vulnerable becomes a complex challenge without accurate knowledge of the underlying cause of the threat, which is the exclusive knowledge of the sorcerer.

Rituals with blessings and chants, and with tobacco, act at the level of recording the underlying cause of the SBE, and not producing a barrier against the snake itself, and are used as rites for the fabrication and transmutations of the person. The use of tobacco deserves to be highlighted due to its common use in protection and healing rituals among the indigenous peoples of the lowlands of South America. Lévi-Strauss points out that tobacco is a food in the spirits' conception, and thus participates in the constitution of the body, providing the ability

to communicate between man and the supernatural order (more intense in the shamans) and also functions as an ontological converter between humans and extra-humans [29,30]. As this author has already pointed out in this study, it was found that other substances share these same properties with tobacco (garlic leaves, bee and wasp hives, resins), '*sucking out*', in the language of the caregiver, the venom inoculated by the snake and rebalancing the sick person's body.

## (Symbolical) Efficacy of the SBE treatment prescribed by the indigenous caregivers

The biomedical healthcare systems focus on the individual in an instrumental disinterested universe; while, in the indigenous system, the individual is part of a collective structure in an intentioned cosmology [31,32]. If the healing is cosmic in the indigenous tradition, in the case of biomedicine, it is chemical [32]. This way of understanding the world reflects on the clinical reality of indigenous societies, and on the way of measuring treatment efficacy. The healing process conducted by the caregivers is imbricated in a structure of trusting relationships between the caregivers, the patient and the community. Thus, it is interesting the way in which the participants of this study emphasize the effectiveness of their practices, and stress their positive results in detailed narratives. In this sense, the ritual's efficacy has three complementary components: the sorcerer's belief in the effectiveness of his techniques; the patient's belief in the sorcerer's power; and the faith and expectations of the community [33]. In addition, the efficacy in healing is largely attributed to the performative aspects of ritual, since the caregivers enact the healing process by calling upon a number of esthetic resources (their chants and prayers), to create a heightened and engaged experience of the participants [31,32].

The attributes for the purpose of healing are generally invoked by the *pajés* from sensitive characteristics of several elements of nature, such as plants and animals. As we observed in this study, the results of the use of a plant or parts of animals are not always expected from its ingestion or topical use, but from its addition to tobacco for smoking or even in rituals. Rituals with chants and blessings include plants and animals that have an attribute considered by the indigenous people as appropriate for the purpose of healing: smell, shape, viscosity, bitterness, sourness, color, and texture [31,34]. The incantations are recited over an intermediate object whose essential function is to provide the incantation with a material support and serve as a vehicle for the therapeutic power, and transfer it to the patient [35]. In this study, bitterness, a crucial characteristic for treatment of SBEs by using plants and bile, is also required in models of efficacy of other ethnic groups, such as those of the Matsigenka [31]. Thus, many bitter plants are used as enchantment vehicles intended to heal wounds of different etiologies, such as SBEs. Although the symbolic properties are of greater interest to indigenous people, these plants commonly produce tannins as secondary metabolites, whose astringent and healing properties could have an effect on the injury caused by snake venom. Moreover, the efficacy of the SBE treatment due to an inhibitory activity on snake toxins cannot be completely ruled out, since many plant-derived molecules have had this ability demonstrated *in vitro* [36]. However, according to our findings, explaining the efficacy of the SBE treatment prescribed by caregivers in the study area in terms of the particular chemical composition of the plants used as vehicles for incantation does not do justice to the indigenous conception of therapeutic efficacy.

It should also be noted that the therapeutic failure, represented by the aggravation of the case, recurrence of symptoms, and even death, is apparently not a reason to question the validity of a treatment method, nor the healing power of a shaman, as already mentioned for the Desana of the Negro River [37,38]. Some SBE cases are caused by serious violations or antipathies (sorcerer-mediated SBEs), or are aggravated by someone not respecting behavioral prohibitions and dietary rules, which set in motion attacks in an intentioned universe characterized

by visible and invisible beings. Contamination by female blood or other fluids is also perceived as a break in the balance between psychic instances or humors and qualities of the body, and weakens patients in recovery, which can make them disabled or even lead to death [25,35]. Furthermore, in many indigenous societies, the tendency is to explain any death that interrupts a person's normal life cycle, before old age, as caused by sorcery [35].

## Boundaries between indigenous and biomedical domains

Within the context of the restructuring of indigenous health policies over the last 20 years in Brazil, and the consequent creation of the Special Indigenous Health Districts from 1999 onwards, the inclusion of indigenous health agents, which are selected among the villagers themselves, appears as a central element of this model [39]. However, contact between different health systems often leads to some kind of conflict or competition [40]. Regarding SBEs, the use of folk medicine and traditional self-care practices are often recorded around the world as the cause of late medical assistance and poor prognosis [41–44]. Our results do not support the simplistic thought that the belief in the efficacy of traditional practices delays the decision to seek the health service in indigenous villagers, as indigenous caregivers are not against transporting patients to the hospital; on the contrary, they recommend referral in these cases. Thus, indigenous caregivers can serve as a point of contact between indigenous and western medicines. The practices carried out by the shamans for the rehabilitation of the SBE patient after hospital discharge are seen as complementary and necessary by the participants. This combination of practices for rehabilitation of SBE patients was also observed in Eswatini, whose main reasons that led patients to resort to traditional caregivers for rehabilitation were the supposedly unsatisfactory treatment in the hospital [28].

The exclusive use of traditional practices may therefore be related to difficulties of access, rather than socio-cultural variables generally considered as "cultural barriers". In fact, this study revealed that indigenous medicine is adaptable to the introduction of biomedical practices and technologies, such as antivenom treatment, postponing the stage of social and body rehabilitation of indigenous people until after hospital discharge. Transfer to hospital actually may have a greater effect on the therapeutic itinerary prescribed by the caregivers than on the biomedical system, in addition to removing the indigenous patient from their community, temporarily undoing affective ties and with the cultural reality. The introduction of indigenous healing practices in the hospital environment could partially resolve this ambiguity. However, conflicting attempts between *pajés* trying to exercise their healing practices in hospitals and health professionals have been recorded in the Brazilian Amazon, with the need for a judicial resolution to ensure the treatment requested by the indigenous patients [45].

The high burden of SBEs among indigenous populations, combined with the structuring of a health system that provides care coverage to many villages, has raised the possibility of decentralizing antivenom treatment to indigenous districts [3]. This would increase access by indigenous groups to proper healthcare, respecting the nexus of these individuals with their territorial, social, and clinical reality. Recently, a care package guideline was validated with the participation of health professionals working in the Amazon region, an essential step for training health professionals that work in indigenous healthcare units [46]. Thus, antivenom treatment would be inserted as a life-saving tool in a world of diverse social, natural, and supernatural representations, in combination with indigenous medical practices.

## Limitations

Regarding the criterion of theoretical saturation of qualitative data, this study probably did not reach complete saturation due to the sample size. This is a sample of an indigenous population

that is difficult to interview due to the need to preserve the identity of the caregivers. During the interviews, the researchers realized that certain rituals were not described in full, and some participants declared that they kept some elements of the ritual secret in order to preserve their culture.

The efficacy of the indigenous medicine is an interesting and complex question. In this work, as much as possible, we avoided classifying the procedures indicated by indigenous caregivers as effective, non-effective or deleterious or judging them in any other way. We believe that it is not up to researchers, at the risk of a colonialist attitude, to make judgments about ancient practices, especially those of a mythical-religious nature. Except for proven deleterious procedures, such as the use of tourniquets, the other practices still lack methodological resources to confirm or refute any benefit to the patient. In addition, we tried our best to respect the role that the caregivers play in their village, including the way they see themselves and the way they prefer to be addressed.

## Concluding remarks

The understanding of SBEs as a sickness in Amazonian indigenous groups of the Alto Solimões River involves the continuous building of the body via proper ritual and behavioral acts, combined with the interaction with other humans and non-humans, including the supernatural ones. In this social and clinical reality, shamans and other indigenous caregivers have the ability to invoke the healing properties of plants, animals, minerals and supernatural entities through the performance of rituals of recognized symbolic efficacy in the villages. Diagnosis and treatment of SBEs are not aimed solely at curing the individual problem, but at converting a biological disorder into a social disorder that is highly mobilizing and that needs to be repaired. It is therefore a matter of analyzing the collective process at stake, which aims to modify or regulate political, economic, or social relations that unite or oppose individuals. The EM presented in this study, although a simplification of a complex clinical reality, in which the SBE represents just one of so many etiological representations of the disorders that affect humans, can serve as a basis for cross-cultural comparisons in the future, as well as for the planning of health actions that truly integrate indigenous and biomedical practices. Our results indicate that the actions of indigenous caregivers do not constitute barriers to the decentralization of antivenom treatment to indigenous health units, in pluralist though combined indigenous-biomedical therapeutic itineraries.

## Supporting information

**S1 File. COREQ Checklist for Qualitative Studies.**
(DOCX)

**S2 File. Transcripts of the interviews.**
(PDF)

## Acknowledgments

We are grateful for the support of the coordinators and professionals who work in the Alto Solimões River Indigenous Health District, who collaborated in the articulation with indigenous caregivers for the collection of data. We would like to thank the technicians of the Health Research Coordination of the National Council for Scientific and Technological Development (COSAU/CNPq) and the National Indigenous Foundation (FUNAI) for providing the permits to carry out this study.

At present, an unprecedented health crisis is ongoing among the Yanomami Indigenous people, a native population in the Brazilian Amazon. We thank all health professionals (those from the Indigenous districts and volunteers), Indigenous leaders and social movements, and Indigenous caregivers, who never abandoned these peoples and who occupy these territories to maintain Indigenous healing practices.

## Author Contributions

**Conceptualization:** Altair Seabra de Farias, Fan Hui Wen, Vinícius Azevedo Machado, Jacqueline Sachett, Wuelton M. Monteiro.

**Data curation:** Altair Seabra de Farias, Felipe Murta, Vinícius Azevedo Machado, Wuelton M. Monteiro.

**Formal analysis:** Altair Seabra de Farias, Felipe Murta, Vinícius Azevedo Machado.

**Funding acquisition:** Felipe Murta, Jacqueline Sachett, Wuelton M. Monteiro.

**Investigation:** Altair Seabra de Farias, Elizandra Freitas do Nascimento, Manoel Rodrigues Gomes Filho, Aurimar Carneiro Felix, Macio da Costa Arévalo, Fabíola Guimarães de Carvalho, Jacqueline Sachett.

**Methodology:** Altair Seabra de Farias, Felipe Murta, Vinícius Azevedo Machado, Jacqueline Sachett, Wuelton M. Monteiro.

**Project administration:** Altair Seabra de Farias, Wuelton M. Monteiro.

**Resources:** Manoel Rodrigues Gomes Filho, Aurimar Carneiro Felix, Macio da Costa Arévalo, Fabíola Guimarães de Carvalho.

**Software:** Asenate Aline Xavier Adrião.

**Supervision:** Fan Hui Wen, Jacqueline Sachett, Wuelton M. Monteiro.

**Validation:** Felipe Murta.

**Visualization:** Altair Seabra de Farias, Asenate Aline Xavier Adrião, Wuelton M. Monteiro.

**Writing – original draft:** Fan Hui Wen, Jacqueline Sachett, Wuelton M. Monteiro.

**Writing – review & editing:** Elizandra Freitas do Nascimento, Manoel Rodrigues Gomes Filho, Aurimar Carneiro Felix, Macio da Costa Arévalo, Asenate Aline Xavier Adrião, Fabíola Guimarães de Carvalho, Felipe Murta, Vinícius Azevedo Machado, Wuelton M. Monteiro.

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
