## [Decision Letter · Decision Letter 0]

24 Jan 2023

Dear Dr Monteiro,

Thank you very much for submitting your manuscript "Building an explanatory model for snakebite envenoming care in the Brazilian Amazon from the indigenous caregivers’ perspective" for consideration at PLOS Neglected Tropical Diseases. As with all papers reviewed by the journal, your manuscript was reviewed by members of the editorial board and by several independent reviewers. In light of the reviews (below this email), we would like to invite the resubmission of a significantly-revised version that takes into account the reviewers' comments. 

We cannot make any decision about publication until we have seen the revised manuscript and your response to the reviewers' comments. Your revised manuscript is also likely to be sent to reviewers for further evaluation.

Sincerely,

Abdulrazaq G. Habib

Academic Editor

José María Gutiérrez

Section Editor

Reviewer's Responses to Questions

**Key Review Criteria Required for Acceptance?**

**Methods**

-Are the objectives of the study clearly articulated with a clear testable hypothesis stated?

-Is the study design appropriate to address the stated objectives?

-Is the population clearly described and appropriate for the hypothesis being tested?

-Is the sample size sufficient to ensure adequate power to address the hypothesis being tested?

-Were correct statistical analysis used to support conclusions?

-Are there concerns about ethical or regulatory requirements being met?

Reviewer #1: -Are the objectives of the study clearly articulated with a clear testable hypothesis stated? Yes

-Is the study design appropriate to address the stated objectives? Yes

-Is the population clearly described and appropriate for the hypothesis being tested? Yes

-Is the sample size sufficient to ensure adequate power to address the hypothesis being tested? Since this is a qualitative study, sample size is adequate.

-Were correct statistical analysis used to support conclusions? No relevant

-Are there concerns about ethical or regulatory requirements being met? None

Reviewer #2: There was a clear study aim and specific objectives; the qualitative design is appropriate; the study population was clearly defined. Given one of the stated study limitations (difficulty achieving data saturation), the sample size is acceptable; the analysis and interpretation of the study results were appropriate. Ethical handling of the study was appropriate.

**Results**

-Does the analysis presented match the analysis plan?

-Are the results clearly and completely presented?

-Are the figures (Tables, Images) of sufficient quality for clarity?

Reviewer #1: -Does the analysis presented match the analysis plan? Yes

-Are the results clearly and completely presented? Yes

-Are the figures (Tables, Images) of sufficient quality for clarity? Yes

Reviewer #2: The analysis presented matched the analysis plan; the results clearly and completely presented; The pictorial presentation results in the figures were beautiful.

**Conclusions**

-Are the conclusions supported by the data presented?

-Are the limitations of analysis clearly described?

-Do the authors discuss how these data can be helpful to advance our understanding of the topic under study?

-Is public health relevance addressed?

Reviewer #1: -Are the conclusions supported by the data presented? Yes

-Are the limitations of analysis clearly described? Yes

-Do the authors discuss how these data can be helpful to advance our understanding of the topic under study? There is a comment on discussion. See the general comment. 

-Is public health relevance addressed? Yes, with in the relevant geographical area

Reviewer #2: The conclusion was appropriate for the results obtained.

**Editorial and Data Presentation Modifications?**

Reviewer #1: (No Response)

Reviewer #2: I will recommend that you ACCEPT for publication in the present form.

**Summary and General Comments**

Reviewer #1: This paper describes the qualitative interview-based study on developing an explanatory model of indigenous healthcare communities for Snakebite envenoming patients from the perspective of indigenous caregivers in selected areas in Brazilian Amazon. The study has been conducted in a well-designed qualitative study method and the presentation of results are appropriate. 

Major comments

More details of the study area: Include a map of the stud area indicating the locations of the caregivers were?

What is the basis of selecting these 8 caregivers and excluding others? Was it due to availability and accessibility or any other reason? Please include the basis of selecting the caregivers. 

In what basis these questions were formulated? Describe the rationale of formulating these questions?

Under data analysis: Lines 225 to 229: “Interview data was analyzed to understand indigenous caregivers’ explanations in an analytical framework with three themes: i) Etiology; ii) Course of sickness (with three subthemes: onset of symptoms, pathophysiology, and severity and prognosis); and v) Treatment”

Describe how the data gathered from listed 20 questions was used to analyse under the above three themes? 

Which questions were related to theme 1, 2 and 3?

Since this manuscript is going to read and refer by the scientists and researchers, highly recommend to include few paragraphs under the discussion to discuss the appropriateness and or inappropriateness of caregiver’s perspectives into western science and medical practices. Eg, use of tourniquet is a dangerous and unaccepted practice in western science and medicine. Likewise, there are many inappropriate and unaccepted practices and believes were explored in this paper. It would be really interesting to discuss these under the discussion. It is understandable and not possible to discuss every finding here. But at least the significant and important findings need to de discuss. 

Other comments

Table 2: Details in the column ‘Presentation”: long descriptions of caregivers are not necessary. Recommend to reduce the word count and present in a point form avoiding sentences. 

239: A total of eight indigenous caregivers were conducted. An important word of “interviews” is missing in this sentence. This has to be edited as “A total of eight interviews were conducted with indigenous caregivers.”

270: SBES: The last “S” should in lower case format. SBEs

263: Agents, should be change as species or snake

Throughout the manuscript: Word Envenoming and Envenomation have been used in different places. Suggest to use one terminology throughout the manuscript.

Reviewer #2: This is a well-written qualitative study, which tenaciously followed the 32-item COREQ checklist.

PLOS authors have the option to publish the peer review history of their article (what does this mean?). If published, this will include your full peer review and any attached files.

Reviewer #1: Yes: Kalana Maduwage

Reviewer #2: Yes: Godpower Chinedu Michael
---

## [Editor Report · Decision Letter 1]

14 Feb 2023

Dear Dr Monteiro,

We are pleased to inform you that your manuscript 'Building an explanatory model for snakebite envenoming care in the Brazilian Amazon from the indigenous caregivers’ perspective' has been provisionally accepted for publication in PLOS Neglected Tropical Diseases.

Best regards,

Abdulrazaq G. Habib

Academic Editor

José María Gutiérrez

Section Editor

---

## [Editor Report · Acceptance letter]

6 Mar 2023

Dear Dr. Monteiro,

We are delighted to inform you that your manuscript, "Building an explanatory model for snakebite envenoming care in the Brazilian Amazon from the indigenous caregivers’ perspective," has been formally accepted for publication in PLOS Neglected Tropical Diseases.

Best regards,

Shaden Kamhawi

co-Editor-in-Chief

Paul Brindley

co-Editor-in-Chief
